# Caring for a Relative with Dementia: Determinants and Gender Differences of Caregiver Burden in the Rural Setting

**DOI:** 10.3390/brainsci11111511

**Published:** 2021-11-15

**Authors:** Dagmar Schaffler-Schaden, Simon Krutter, Alexander Seymer, Roland Eßl-Maurer, Maria Flamm, Jürgen Osterbrink

**Affiliations:** 1Institute of General Practice, Family Medicine and Preventive Medicine, Paracelsus Medical University, 5020 Salzburg, Austria; maria.flamm@pmu.ac.at; 2Institute for Nursing Science and Practice, Paracelsus Medical University, 5020 Salzburg, Austria; simon.krutter@pmu.ac.at (S.K.); roland.essl@pmu.ac.at (R.E.-M.); juergen.osterbrink@pmu.ac.at (J.O.); 3Department of Sociology, Paris Lodron University, 5020 Salzburg, Austria; alexander.seymer@plus.ac.at

**Keywords:** caregiver burden, caregiver gender, dementia care, family caregivers

## Abstract

Dementia is a progressive disease that puts substantial strain on caregivers. Many persons with dementia (PwDs) receive care from a relative. Since male and female caregivers experience different issues related to stress, it is important to meet their different needs to prevent the early nursing home placement of PwDs. This study investigated the multifactorial aspects of caregiver burden and explored gender differences in caregiver burden in a rural setting. This was a cross sectional study that administered anonymous questionnaires to family caregivers of PwDs. Caregiver burden was assessed using the Burden Scale for Family Caregivers—short version. A path model was used to determine the multivariate associations between the variables. To reflect the multifaceted aspects of caregiver burden, we used Pearlin’s model with its four dimensions. A total of 113 family caregivers responded to our survey. The overall burden of caregivers was moderate. According to the path model, gender differences were predictors of caregiver burden. The behaviour of the person with dementia and cohabitation had direct effects on caregiver burden. Our results suggest that the experiences of men and women caring for a PwD are different and highlight the need for tailored support in dementia care.

## 1. Introduction

The increasing number of aging people in western countries is one of the most significant economic, social, and medical issues of current times [1]. The European population is aging particularly fast as a result of prolonged life expectancy and a decline in birth rates [2]. Aging is a risk factor for the vast majority of chronic diseases and the strongest known driver of dementia [3]. Approximately 46.8 million patients are currently diagnosed with dementia worldwide, and by 2050, this number is expected to rise to 131.5 million [4].

Due to an irreversible decline in cognitive, social, and physical function, persons with dementia (PwDs) can develop a need for care in the early stages of the disease and become increasingly dependent on caregivers as the disease progresses [3]. The majority of PwDs live in the community, where relatives play an essential role in the maintenance of care at home, delaying transition into formal care institutions, and contributing to the overall quality of life of the care recipient [5]. In most cases, one member of the family (e.g., spouse, daughter, or son) assumes the role of the main care provider, often supported by formal care services. Due to demographic and socio-cultural changes, adult children often do not live together with their parents. Hence, many spouses have to take over care responsibilities for a PwD without the support of the family. Due to neuropsychiatric and behavioural symptoms, caring for a person with dementia is particularly stressful, and the risk of caregiver burden is higher compared to caring for a person without dementia [6]. Compared to people of the same age and living circumstances, caregivers of PwD are more likely to report depression, rate their health status lower, and take more medication [7].

### 1.1. Family Caregiver and Caregiver Burden

In the literature, the term family caregiver is used to encompass all informal caregivers (e.g, family members, friends, and neighbours) of a PwD who provides support [8]. The domains of the caregiving role include assistance with household tasks, self-care tasks, and mobility; provision of emotional and social support; health and medical care; advocacy and care coordination; and surrogacy (e.g., handle financial and legal matters) [9]. Given this wide array of role expectations, the concept of care providers and care managers is very useful in understanding the caregiver role [10]. Family caregiving for a PwD therefore cannot be defined as merely the performance of tasks and assisting in daily practical activities. Moreover, a family caregiver also emphasizes their concerns, meanings, and purpose of caring for the PwD when describing their role [11]. In Nolan’s holistic typology, family caregiving therefore extends beyond hands-on care and encompasses anticipating the future support needs of the PwD, monitoring and supervising, preserving the PwD’s sense of self, and helping the PwD to develop new and valued roles [12]. As dementia is a progressive disease, the caregiving role is shaped and expanded by the transitions in dementia caregiving over time. These transitions and role changes, along with the behavioural, health, and functional status of the PwD, affecting the social, physical, and emotional health of the caregiver over time [9]. Furthermore, many caregivers of PwD are older themselves, physically frail, and have health problems of their own [8].

Even though recent research emphasises the positive aspects of family caregiving and has shown caregivers experience personal growth, satisfaction, a sense of fulfilment, and feeling good about caring [13], the challenges of caring for a PwD are often burdensome. In this context, the term caregiver burden refers to an experience affecting caregivers’ physical and emotional health and their social life throughout the progressive course of the disease [14]. Resistance to care, aggression, and disruptive behaviour of the PwD are challenging for family caregivers, who usually lack training and professional skills to fulfil their responsibilities [15].

Several determinants associated with the development of caregiver burden have been identified: Characteristics related to the PwD (e.g., disruptive behaviour, duration of dementia, and functional status) and caregiver-related factors (e.g., cultural dimensions, the relationship with the PwD, cohabitation, and coping strategies) can have a significant impact on the experienced strain of family caregivers in dementia care [16]. Studies suggest a multifactorial role and interdependencies of these determinants of burden in dementia care [17,18].

Pearlin et al. [18] developed an extensive model of caregiver stress [17]. It is assumed that caregiver burden is a mix of circumstances, experiences, responses, and resources with variable impact on caregivers’ health and behaviour. Four major domains that contribute to caregiver burden are described in the model: First, the background and context of the stress (key characteristics of caregiver, type of dyadic relationship, caregiving length, level of support, and impact of other life events on the ability to cope). Second, the primary stressors of the illness (the patients’ cognitive ability, the level of help and surveillance, and behavioural and psychological symptoms). The third area covers the secondary role strains (occupation, economic problems, family conflict, relationship quality, and social life outside of the caregiver role). The fourth area considers the intrapsychic strains of the caregiver (personality, level of confidence and competence, and role captivity experienced by the caregiver) [17,18].

A recent study with a large sample revealed that the median time from diagnosis of demen tia until institutionalisation of the PwD was 3.9 years [19]. Hence, it is important to identify caregiver burden in time to prevent caregiver overload and early nursing home placement of the PwD, who have a manifold higher hazard for institutionalisation than care recipients without dementia [20].

### 1.2. Gender and Caregiver Burden

Traditionally, in most cultures, the vast majority of family caregivers are women, but men are assuming caregiving roles at an increasing rate, making up almost 40% of caregivers in Canada and the United States [21]. In the future, caring duties will increasingly be carried out by husbands and sons of PwDs, as the number of women with dementia increases and daughters are more engaged in the workforce or live farther away [22]. Although there is extensive literature on informal caregiving in dementia, only a few studies have highlighted the caregiving role of men [23,24,25]. In particular, studies on the caregiving role of sons are scarce [26]. Regarding the projected increasing number of male caregivers, it is important to explore the differences in how men and women are coping with the challenges of caring for a PwD [27]. Many studies exploring gender differences in caregiving have reported a higher burden in women, although the results are inconsistent [28,29,30]. Several explanations can be considered for this imbalance. Female caregivers tend to experience higher levels of physical and mental strain and often provide care for a longer period compared with male caregivers [31,32]. Male family caregivers tend to see caregiving tasks less emotionally than female family caregivers, which may contribute to reducing caregiver strain [33]. In contrast, men are likely to seek formal care support less often [34]. Traditional gender roles still expect women to take over caring and family obligations, while men focus on their careers. This gender role conflict may place a burden on caring for men and prevent them from seeking support, whereas women may feel obliged to fulfil their expected role [35].

Many studies on family caregiving have included gender as an independent variable, but few have approached gender as a source of individual differences [36]. Moreover, gender is often simplified as a comparison between men and women, without considering individual differences. Gender equality in caring for relatives is probably still far away in most countries, and women experience disadvantages due to lower income and poorer education [22]. However, it seems important to further investigate how gender operates as one of the multiple factors that lead to caregiver burden.

### 1.3. The Rural Setting

Numerous studies have addressed the stress experienced by family caregivers of PwDs [16]. To date, most studies on family caregivers have been conducted in urban settings [37,38]. It is still unclear whether caregivers of PwDs experience more stress in rural areas than in urban areas. Caregivers in rural areas often experience barriers to professional health services. It is therefore not surprising that they also have more informal support by other family members compared to urban caregivers [39]. Several studies indicate that female and male caregivers experience different types and levels of burdens in rural areas [37,40]. As in most studies about caregiver burden, women and spouses caring for a PwD reported a higher burden compared to men in rural settings [37]. Older age and the female gender are predictors of service utilisation when caring for a PwD in a rural setting [41]. A Swedish study reported financial concerns of rural caregivers of a PwD compared to urban caregivers. The authors assumed this result is probably a gender issue because most rural caregivers in the study were female spouses [42]. In another study, male caregivers of a PwD in urban areas reported better mental health than their counterparts in rural areas [43]. However, study results regarding caregivers of a PwD in rural and urban settings are conflicting and sample sizes are small.

The aims of this study were (1) to investigate the caregiver burden of family caregivers of PwDs, (2) to explore the multifactorial aspects contributing to caregiver burden, and (3) to investigate the gender differences in the experience of burden in a rural setting.

## 2. Materials and Methods

The data for this cross-sectional study were collected as part of the interdisciplinary PAiS project (Pflegende Angehörige von Menschen mit Demenz in Salzburg—Family Caregivers of Persons with Dementia in Salzburg), and focused on the caregivers of people with dementia in rural-provincial areas of Salzburg, Austria. The project applied mixed methods, including surveys of family caregivers of PwDs, professional home care nurses, general practitioners, and qualitative interviews with caregivers (for details see [44]). For this study, we analysed the quantitative data of the survey of family caregivers.

### 2.1. Sampling

The postal survey of family caregivers of PwDs in the rural area of Salzburg county was conducted between November 2016 and May 2017. The inclusion criterion was care for a PwD at home in a rural area of Salzburg county. Salzburg county, excluding the city of Salzburg, can be defined as intermediate or thinly populated according to the European Commission [45]. As the study focused on rural areas, caregivers of PwDs with a residence in the city of Salzburg were excluded from the study.

The sample was recruited using a combination of non-random procedures to address a larger sampling frame. First, home care nurses and general practitioners promoted the survey in their daily work with caregivers. Second, the study was advertised in mass media, regional newspapers, and local trains. Third, information material and flyers were distributed in day care centres, pharmacies, acute geriatric facilities, meals on wheels, and support groups for family caregivers. All three channels provided contact information, inviting caregivers to participate in the survey or to contact the study team. Eligible family caregivers who were interested in participating received a self-administered paper-based questionnaire (PAPI) with a free return envelope. Submission of the questionnaire was considered consent.

### 2.2. Instruments

The 43-survey items covered care arrangements, usage, and knowledge of local healthcare services, caregiver burden, and the demographics of the caregiver and the PwD. To systematically explore the multifactorial aspects contributing to caregiver burden, independent variables in this study were based on Pearlin’s model of caregiver stress and its four dimensions (see Figure 1) [18]. Following the model, we considered the background and context of caregiver burden in relation to the age and gender of the caregiver, type of relationship and care duration, primary stressors with functional activities of daily living (ADLs), dependence (Barthel Index [46]), and disruptive behaviour of the PwD (Nurses’ Observation Scale for Geriatric Patients (NOSGER [47])); secondary role strains with employment status, income, and cohabitation; and intrapsychic strains with the caregiver’s quality of life (EUROHIS-QOL [48]).

Caregiver burden was assessed using the Burden Scale for Family Caregivers—short version (BSFC-s) in accordance with Graessel et al. [49]. The instrument measures reduced life satisfaction, physical exhaustion, wanting to run away, depersonalisation, decreased standard of living, health affected by caregiving, caregiving taking strength, conflicting demands, future worries, and impact on social relationships on a 4-point scale, with responses ranging from 0–3 (0 = ‘not true’, 1 = ‘a little true’, 2 = ‘mostly true’, and 3 = ‘true’). The total score is determined by summing all 10 items and is used to assess the subjective burden of informal caregivers. Scores below 5 indicate no or only a minor burden. Scores between 5 and 14 represent a medium level of burden. A score of 15 to 30 indicates a severe to very high level of burden [50].

The NOSGER [47] is an instrument that captures the daily behaviours in 2 weeks of psychogeriatric patients. The instrument covers six dimensions: Memory, instrumental activities of daily living, self-care, mood, social behaviour, and disruptive behaviour. There are five items for each dimension, for a total of 30 items. Each of the 30 items is measured using the same 5-point scale, ranging from 1 to 5 (1 = never, 2 = sometimes, 3 = often, 4 = mostly, and 5 = always). The total score is the sum of the scores of the five dimensions and ranges from 5, corresponding to never showing the specific behaviour, and a maximum of 25, indicating permanent behaviour. In the PAiS project, we used all 30 items. For the study presented here, only the item ‘Disruptive behaviour’ was used, as previous studies with the data [41] indicate that this item (Cronbach’s alpha = 0.662) is highly relevant for the situation of family caregivers. The total score reflects the frequency of disruptive behaviour with a range of 5 to 25.

The Barthel Index is a measure of independence in ADL, covering 10 different activities: Eating (scale: 0, 5, 10), bathing (scale: 0, 5), washing (scale: 0, 5), dressing (scale: 0, 5, 10), bowel control (scale: 0, 5, 10), bladder control (scale: 0, 5, 10), getting on and off the toilet (scale: 0, 5, 10), moving out of bed and from wheelchair to bed (scale: 0, 5, 10, 15), stair climbing (scale: 0, 5, 10), and walking (scale: 0, 5, 10, 15). The categorical items comprise the sum of the Barthel Index and indicate full independence with the highest category for each item and a maximum of 100 on the index and complete dependency with the lowest categories on each item summing up to 0 in the index (Cronbach’s alpha = 0.903) [46].

The EUROHIS-QOL is an 8-item self-assessment of quality of life that covers life satisfaction, satisfaction with health, having energy for everyday life, satisfaction with the ability to perform daily activities, self-satisfaction, satisfaction with personal relationships, having enough money, and living conditions. All items are based on a 5-point Likert scale, and the total score is the sum of the eight items with higher scores indicating higher quality of life (Cronbach’s alpha = 0.839) [48].

Care duration was captured in months or years in the survey and used as the duration in years in the analysis. We collected data on part-time employment with a question on employment status with full-time, part-time, housewife/man, retired, in training, or unemployed as response options. Cohabitation was determined with a yes/no question on living in the same house or flat. Education was measured as the highest level of education of the caregiver, with the answer options of compulsory school, apprenticeship, technical or commercial school, high school, or a (technical) college/university.

### 2.3. Analysis

The analysis was conducted with R version 4.0.5 [51] and most of the results are presented as three-way plots, including bivariate parametric and non-parametric tests for mean differences. Pairwise deletion was applied because of the small sample size. The small subsample size of male caregivers limited the possibility of more elaborate multivariate statistical analyses. Nonetheless, a path model was estimated to provide some direction for multivariate associations. The conclusions from the path model were exclusively descriptive.

## 3. Results

The study population included 113 family caregivers, of which 27 were men (23.9%). Most caregivers in our study were children of PwDs (45.9%) and lived in the same household with the PwD (61.1%). In total, 50% of men and 38.1% of women were employed, and female caregivers were mostly working part time. Approximately half of the caregivers had a low level of education (max. apprenticeship), and a low monthly net income (up to 2000 €). The PwDs in our study population had a relatively high independency in their ADLs (66.5 mean on the Barthel Index) and a low level of disruptive behaviour (10.2 mean on the NOSGER scale). Most of the caregivers used home care nursing services (61%). Male and female caregivers showed no differences in quality of life (EUROHIS-QoL). These scores are comparable to the EUROHIS-QoL levels of the general population [48].

Regarding gender differences in our sample, there were no or only marginal differences between male and female caregivers with respect to the age of the caregiver, relationship to the PwD, education, cohabitation, or disruptive behaviour of the PwD. Male caregivers were slightly more often married, had a higher monthly net household income, were not working part time, and lived in more densely populated areas than female caregivers (see Table 1). Men provided care for a significantly shorter overall time. They cared for a PwD for a similar number of days per week, but spent fewer hours per day on caregiving tasks compared with women. Finally, male caregivers used home nursing services more often. Irrespective of the caregiver’s gender, male PwDs had a minimum Barthel score of 40, whereas female PwDs had a minimum score of 0.

Overall, our study population reported a moderate burden, albeit on the border to severe stress (13.6 on the BSFC-s, see Figure 2). Our data indicated gender differences in burden. Female caregivers scored higher on the BSFC-s than male caregivers. Although men showed a normal distribution around the mean burden, women were equally spread over the entire burden scale. Hence, the group of male caregivers in our sample was more homogenous than the very heterogeneous group of female caregivers with respect to caregiver burden (see Figure 2). Overall, male caregivers showed a lower burden (see Figure 2) and care for a PwD with higher independence in ADLs.

Considering the differences in the univariate analysis between male and female family caregivers, several associations between caregiver burden and some characteristics deserve further attention.

In the analysis of caregiver burden, the results indicate a clear difference between men and women. For male caregivers, cohabitation had only a marginal and insignificant impact on the level of burden. In contrast, female caregivers living with a PwD had a significantly higher BSFC-s score compared with female caregivers living in separate households. The larger spread of women across the BSFC-s scale pronounces the difference between cohabitation and no cohabitation in comparison of the medians. Women living with a PwD had a significantly higher risk of caregiver burden (see Figure 3).

In the analysis of the type of dyadic relationship, a more complex relationship was found between caregiver burden and the kinship of the PwD. Overall, caregiver burden was higher for spouses than for children. Male PwDs as spouses or partners were found to be associated with the highest caregiver burden, with means around 18 and medians above 20 on the BFSC-s (*F* = 16.71, *df* = 2, *p* = 0.000). Among female PwDs, spouses, partners, and mothers were associated with a similar level of burden (*F* = 1.82, *df* = 2, *p* = 0.179). Since male caregiver burden was insignificant when comparing the relationship with a PwD (*F* = 2.33, *df* = 2, *p* = 0.240), the mean burden for spouses was also around 13. For female caregivers, the burden differed significantly (*F* = 9.23, *df* = 2, *p* = 0.000), with spouses causing the highest. Female caregivers caring for a parent had an average BSFC-s score of around 13. All other combinations had mean values below 10.

With regard to employment, female caregivers working part-time had a higher burden than full-time caregivers. There were no part-time working male caregivers in our sample. In general, working caregivers had a lower burden than caregivers who were retired (see Appendix A)

One of the most striking gender differences was the duration of care. Figure 4 compares the association between BSFC-s and duration of care in years between male and female caregivers. The plots illustrate two key differences. No males reported a duration of care longer than 5 years. Most women reported a duration of care of up to 7 years, but the data showed multiple entries with significantly longer care durations. Although the burden for men was unrelated to the duration of care, we found an almost linear relationship with increasing burden for up to 7 years of care for female caregivers. No PwD who received care from a male caregiver received a care allowance higher than level 5 (levels 6 and 7 indicate a higher dependency of a PwD).

A similar result was found in the comparison between caregiver burden and disruptive behaviour of the PwD; while for male caregivers, the correlation between BSFC-s and disruptive behaviour was not significant, female caregivers showed a strong link between disruptive behaviour and caregiver burden (see Figure 5).

Since the results drawn on bivariate relationships grouped by gender and sample size limited the possibility for a multivariate analysis of the data, we designed a path model to obtain a better understanding of the associations in the sample. The path model in Figure 6 reflects the understanding that gender has no direct impact on caregiver burden, but gender differences as predictors of caregiver burden account for it. In methodological terms, the gender effect is fully mediated by different covariates, namely disruptive behaviour, part-time employment, care duration, gender of the PwD, and cohabitation. The model included 72 cases and fit the data well (comparative fit index = 0.992, root mean square error of approximation = 0.026, SRMR = 0.053, Bayesian information criterion = 1523.029, R-Square for the BSFC-s was 0.173; [52,53,54]).

The model provides four key insights: First, the direct effect of the mediating predictors on caregiver burden (see the arrows towards BSFC-s on the right side of Figure 6); second, how male and female caregivers differ on the mediators (see grey boxes in Figure 6); third, how gender acts as a predictor for the mediators compared with age (white boxes below the grey boxes in Figure 6); and fourth, the combination of the direct effects on the mediating predictors and their impact on the BSFC-s score provides insights about indirect gender differences in caregiver burden.

With respect to the direct effects, cohabitation and disruptive behaviour were the only statistically reliable effects on caregiver burden. Taking into account the mean BSFC-s score of 13.6, cohabitation alone increased the BSFC-s score by an average of 3.649 points. Considering the range of 12 for disruptive behaviour, the unstandardised effect of 0.721 indicated a similar magnitude. Although the *p*-value for cohabitation (0.079) can only be interpreted as a trend, the effect of disruptive behaviour increasing the caregiver burden is statistically significant at the 0.05 level. Working part time, duration of care, and the gender of the PwD were not significant, but have sizeable effects on caregiver burden.

The gender of the caregiver showed a significant association with part-time employment (0.316), duration of care (1.847), and gender of the PwD (−0.277). Women were more likely to work part-time and have longer durations of care. There were no differences between male and female caregivers in terms of cohabitation and disruptive behaviour. As gender impacted only predictors that had no significant direct effect on caregiver burden, the total effect of gender on caregiver burden also lacked significance (*b* = 0.995; *p* = 0.418).

Caregiver age significantly affected part-time employment (−0.011), cohabitation (0.017), and the gender of the PwD (−0.011). Older caregivers more often cared for a male PwD. Although the unstandardised effects are smaller for age, considering the range for age and the significance of the total effect (*b* = 0.084; *p* = 0.045), the relevance of caregiver age is higher than caregiver gender for the subjective caregiver burden.

In terms of indirect gender effects, the model provided evidence of several significant direct effects, but no single path from gender to BSFC-s score was significant. For both the mediators with a direct effect on BSFC-s score—cohabitation and disruptive behaviour—gender was not a significant predictor. The direct impact of gender on the mediators confirmed previously observed associations, such as women are more likely to work part-time or spend more time caring for the PwD.

## 4. Discussion

This study examined the burden experienced by family caregivers of PwDs in rural settings and explored the effects of gender on caregiver burden. Overall, the level of burden was somewhat similar between male and female caregivers in our study, but female caregivers perceived a slightly higher strain and several predictors of burden showed differences between genders.

### 4.1. Gender Aspects

Our results were in line with those of previous studies suggesting that women generally experience higher caregiver burden when caring for a PwD compared with male caregivers [37,55,56]. This is particularly true when the duration of care is longer [57]. Female caregivers have also reported higher emotional distress and poorer physical health than their male counterparts [16]. The higher burden on female caregivers may be attributed to secondary stressors, such as financial or family problems [58]. The commonly reported lower burden of male caregivers may also be the result of traditional gender roles. Men tend to conceal their perceived caregiver strain to avoid being considered weak [33]. As in many other countries, responsibility for care in Austria traditionally lies with women. Caregiving as an obligation and family responsibility creates stress, especially for spouses [59]. It also seems that female and male caregivers perceive different caregiving issues as stressful. Men seem to have a different approach to caregiving than women. They tend to be more problem-orieorntated in their caregiving and want to control the situation, whereas women react more emotionally, which increases perceived caregiver strain [23]. Our results also showed that women spend more hours per day caring for PwDs. They tend to give up their caregiver role later than men and receive less formal support [59].

### 4.2. Effects of Kinship and Cohabititation

Similar to other studies, living with a PwD was found to increase caregiver burden [59,60]. This was highly significant for the women in our study. Women provide care for more hours per day and therefore tend to neglect their social lives. A 24-h obligation is a major risk factor for burden in dementia care and may also lead to overload [59,61,62]. Moreover, for female caregivers, the PwD was most often a male spouse [6]. In our study population, male spouses were linked to the highest caregiver burden. The gender of the PwD seemed to play an additional role in caregiver burden. Men caring for a female PwD experienced less burden compared with female caregivers caring for a male PwD [63]. This may be due to the fact that most male PwDs receive care from their spouse, which in turn results in higher burden. Spouses may have more stress because their relationship with a PwD involves more than only providing care. Since spousal caregivers are usually older than other family caregivers (e.g., daughters), they often have health-related problems of their own, and the duration of care is longer. As dementia progresses, emotional stress increases, and caring spouses report reduced satisfaction with their relationship and quality of life [64]. Spousal caregivers of a PwD participate less often in social activities than other family caregivers, which is also associated with increased caregiver burden [59]. Although caring for a spouse is particularly stressful, several authors have reported a higher burden in caregiving adult children [28,65]. Children may perceive caring for their parents as an additional strain, making reconciliation of their job and family commitments more difficult [66]. Women generally often have part-time jobs and, therefore, experience multiple burdens due to working, caregiving, and family obligations [67]. As daughters are more likely to care for a parent with dementia compared with sons, caring adds another dimension to the multiple social roles, such as parenting, employment, and marital life faced by women. Caregiving sons receive more informal assistance in their caregiving roles, especially from their wives [27]. Conflicts among siblings over caregiving issues also frequently cause stress for adult children caregivers [59].

### 4.3. Employment

Employment may have positive or negative effects on caregivers. An important factor is the flexibility of the employer and the caregiver’s ability to balance their work and caregiving roles [68]. However, working caregivers reported less burden compared with retired caregivers in our study. Usually, working caregivers are younger and are more financially able to afford formal care services. Several authors have reported that paid employment outside the home may have beneficial effects for the caregiver [22]. Working caregivers experience less burden and role overload due to better income and more social interaction. Moreover, paid employment can ease the financial strain [69]. Caregivers who work full time tend to use formal care services more often, which may also contribute to the relief of burden. This might have been the case in our study, as 50% of male caregivers worked full-time and 78% used home care services. Again, in Austria, men are more often engaged in paid work than women who stay at home due to housekeeping and family obligations such as caring for family members.

### 4.4. Care Duration and Dependency of the PwD

We observed a significant gender difference in duration of care. Duration of care is a relevant predictor of caregiver burden and does not necessarily depend on the severity of the PwD’s condition [57]. Due to the progressive character of dementia, the PwD becomes increasingly dependent over time, while caregivers grow older and likely develop their own health problems. Our results clearly indicate that caregiver burden increased with the duration of caregiving in women, whereas men showed a constantly high level with ongoing duration. Additionally, the overall duration of care for male caregivers was significantly shorter than that of female caregivers. As a result, men in our study population did not care for a PwD receiving the highest care allowance (those with worse health conditions) and the PwD had a lower level of dependency in ADLs compared with PwDs of female caregivers. There are several possible explanations for this result. The majority of male caregivers in our study worked full time, so it is conceivable that the increasing dependency of the PwD impacted their ability to work full time and might have led to the early institutionalisation of a PwD. On the other hand, women tend to use formal homecare more often [41], which enables the caregiver to maintain the care arrangement at home for a longer time. Furthermore, female spouses are often younger than the PwD and can provide care for a longer time [31].

### 4.5. Service Utilisation

The level of support is another important dimension affecting gender differences in caregiver burden. In our study, male caregivers used formal care services more often than women (78% vs. 55%). Regarding service utilisation, several authors reported lower caregiver burden and depression in male caregivers due to higher service utilisation [58]. Other authors have reported that men are more likely to decline formal support than women [23]. Reluctance to transfer their caring responsibilities to others can also be a barrier for men using formal care services [70]. In contrast, a recent systematic review found that male caregivers tend to seek help earlier than female caregivers, who reject professional support more often due to feelings of obligation and guilt [59,71]. Lack of information or ambivalent attitudes towards services are often mentioned as reasons for non-utilisation. However, regarding help-seeking patterns, gender-related results are controversial [72].

### 4.6. Path Model and Pearlin’s Four Dimensions Contributing to Caregiver Burden

To explore the multifactorial aspects that contribute to caregiver burden, we performed a path model analysis. Our path model showed that gender has no significant direct impact on caregiver burden, but that gender differences in predictors of caregiver burden account for it. The path model revealed that caregiver age had a stronger direct effect on caregiver burden than gender.

According to Pearlin’s extensive model of caregiver stress and its four dimensions [18], the primary stressors of the illness play a key role; disruptive behaviour of the PwD is one of the strongest contributors to caregiver burden. Behavioural and psychological symptoms are the most common causes of institutionalisation of PwDs [73]. The decline in cognitive and functional abilities also leads to a significant increase in caregiver burden [74]. This seems plausible because aggression and delusion are particularly stressful for caregivers [75]. In our participants, disruptive behaviour was significantly related to burden, but only for women after adjusting for gender. This confirms the results of previous studies suggesting that the challenging behaviour of a PwD is more problematic for female caregivers than for male caregivers [76]. The other primary stressor in our model, the dependence in ADLs of the PwD, did not contribute to caregiver burden in our path model.

The second largest direct contributor to caregiver burden in our path model was cohabitation. Cohabitation has been found to be a risk factor for caregiver burden [16] and this burden increases for spouses over time [77]. Cohabitation may also contribute to the stressful feeling of role captivity [17].

With regard to the background and context of the stress as outlined in Pearlin’s model, in our study the age of the caregiver, the gender of the PwD, the dyadic relationship, and the duration of care had an impact on caregiver burden. Older caregivers are mostly retired, live in cohabitation with the PwD, and often have health-related problems, and the duration of care has persisted for a longer period of time [64,77]. Older caregivers care more often for male PwDs. A recent systematic review showed that, particularly for women, the burden increases over time in dementia care [77].

The fourth domain in Pearlin’s model, intrapsychic strains, was not considered in our path model as a predictor variable, as our sample indicated gender differences in QoL. Male caregivers tended to have a higher QoL than female caregivers [78]. In sum, our results suggest that care duration, disruptive behaviour, and role captivity, as intrapsychic strains are the main predictors of caregiver burden in caregivers of PwDs, particularly for female caregivers.

### 4.7. Strengths and Limitations

This study has several limitations. The study sample was a non-randomised sample of rural areas in Salzburg county with a certain probability of sampling bias. Most of our caregivers used home care nursing services, which was most likely a result of the recruiting strategy. As this study was conducted in a small part of Austria, the results should be interpreted with caution. The generalisability of the study results is therefore only possible to a limited extent as data collection was conducted in rural areas of Salzburg county. A classical estimation of the sample size was not possible because no valid data on the number of dementia patients in the study area is available. Due to the recruitment strategy, we do not have any information about refusals or the number of withdrawals. In our study, male caregivers were slightly less burdened and lived in more densely populated areas than female caregivers. Nevertheless, our results do not allow us to draw any conclusions about whether caregivers of PwDs in rural areas are more burdened because we lack data from urban settings as a comparison. Additionally, we failed to carry out in-depth qualitative interviews with male caregivers using our mixed-methods approach. The results were based on the quantitative data of our study.

Despite these limitations, the present study has several important strengths. To our knowledge, this is the first study to explore the gender-specific aspects of caregiver burden in dementia care in Austria and one of the few in a rural provincial area. Furthermore, the use of the path model and the interpretation of our results, which were based on Pearlin’s extensive model of caregiver stress and its four dimensions, offered a deeper understanding of the associations between the predictors of the complex phenomenon of caregiver burden in dementia care.

## 5. Conclusions

Caregiver burden is multifaceted and cannot be simply reduced to gender-related differences. With an increasing number of women diagnosed dementia, it will be more important to understand the gender-specific needs of caregivers to provide suitable health care services and support. Male caregivers have different needs than female caregivers in terms of support. It is essential to identify early predictors of burden to protect the caregiver from health problems and prevent the institutionalisation of PwDs.

Our findings suggest that the experiences of men and women caring for a PwD can be very different, despite apparently similar circumstances, and highlight the need for formal and informal support tailored to the specific needs of men and women and their gendered role in dementia care.

## Figures and Tables

**Figure 1 brainsci-11-01511-f001:**
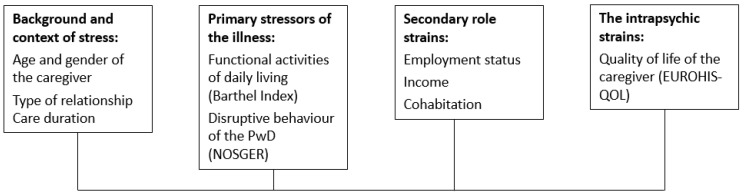
Pearlin’s extensive model of caregiver stress, the four major domains that contribute to caregiver burden and the independent variables used to measure each dimension.

**Figure 2 brainsci-11-01511-f002:**
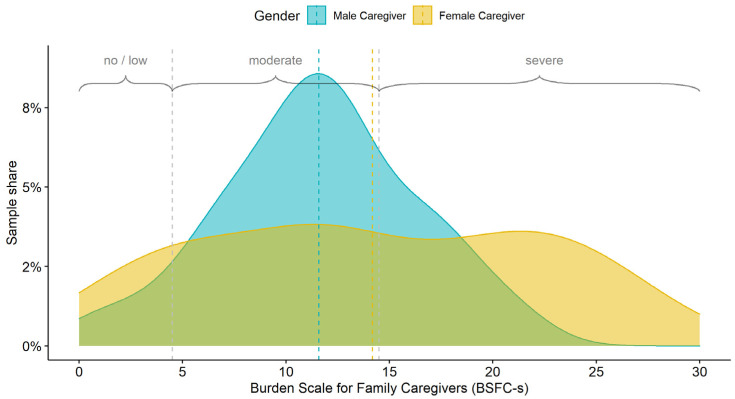
Distribution of Burden Scale for Family Caregivers—short version (BSFC-s) score by gender.

**Figure 3 brainsci-11-01511-f003:**
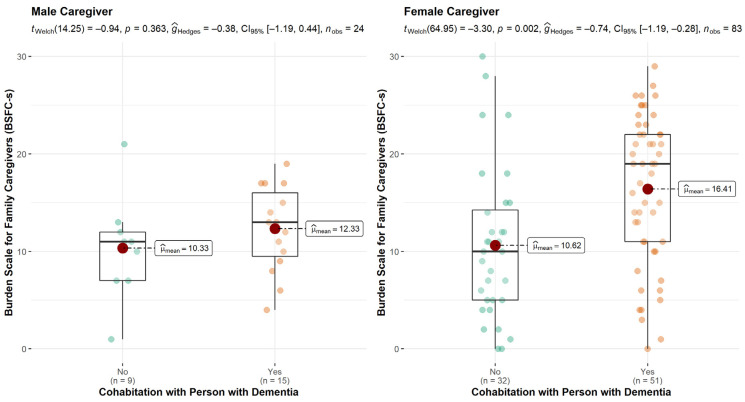
Association between caregiver burden and cohabitation by gender of caregiver.

**Figure 4 brainsci-11-01511-f004:**
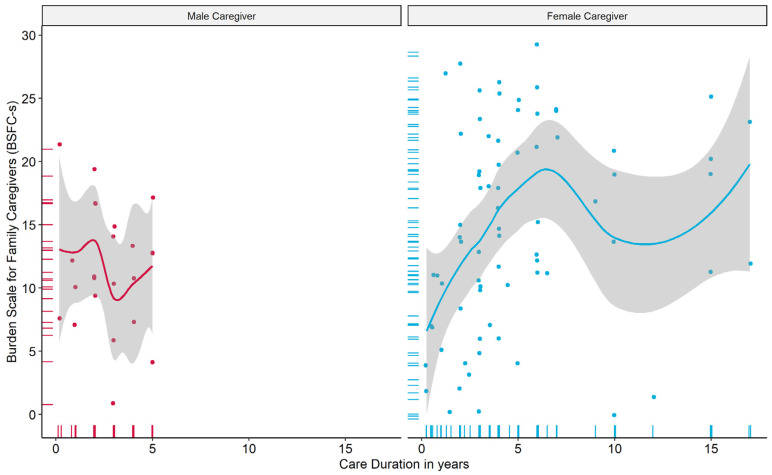
Association between caregiver burden and care duration by gender.

**Figure 5 brainsci-11-01511-f005:**
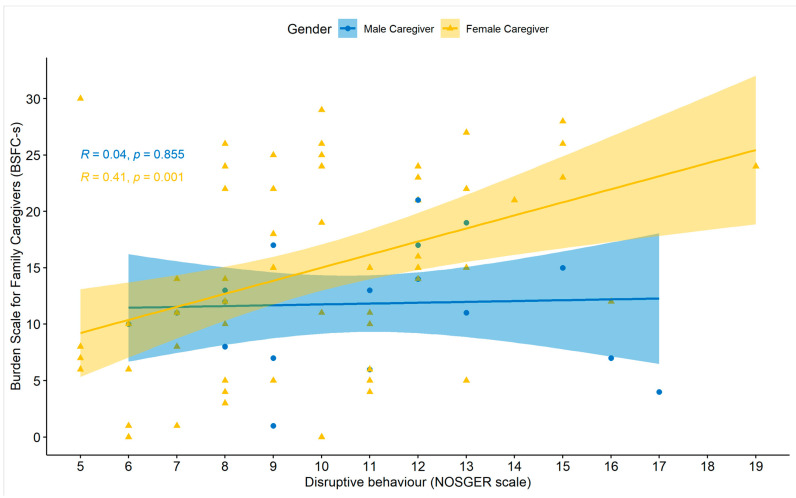
Association between caregiver burden and disruptive behaviour by gender. *BSFC-s =* Burden Scale for Family Caregivers—short version, *NOSGER =* Nurses’ Observation Scale for Geriatric Patients.

**Figure 6 brainsci-11-01511-f006:**
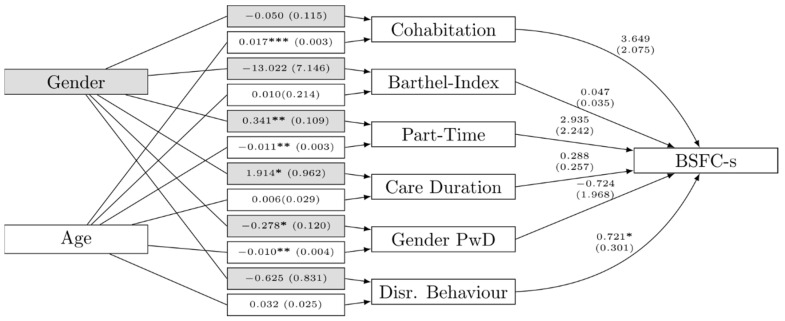
Path model for caregiver burden. Note: Bold figures indicate significant coefficients on a 0.001 level (***), a 0.01 level (**), or a 0.05 level (*). Italic figures indicate a trend with a *p*-value < 0.1.

**Table 1 brainsci-11-01511-t001:** Sample characteristics.

	Male Caregiver (*n* = 27)	Female Caregiver (*n* = 86)	Overall (*n* = 113)	*p*-Value
Age (years)				
Mean (SD)	61.4 (18.4)	60.5 (13.8)	60.7 (14.9)	0.830 ^1^
Relationship with PwD				
Spouse or partner	10 (37.0%)	25 (29.8%)	35 (31.5%)	0.803 ^2^
Mother/father	14 (51.9%)	37 (44.0%)	51 (45.9%)	
Sister/brother	0 (0.0%)	2 (2.4%)	2 (1.8%)	
Daughter/son	1 (3.7%)	6 (7.1%)	7 (6.3%)	
Daughter/son in law	0 (0.0%)	2 (2.4%)	2 (1.8%)	
Grandmother/grandfather	1 (3.7%)	6 (7.1%)	7 (6.3%)	
Friend	1 (3.7%)	6 (7.1%)	7 (6.3%)	
Marital status				
Married/cohabitation	22 (81.5%)	62 (72.1%)	84 (74.3%)	0.218 ^2^
Separated/divorced	2 (7.4%)	17 (19.8%)	19 (16.8%)	
Widowed	0 (0.0%)	3 (3.5%)	3 (2.7%)	
Single	3 (11.1%)	4 (4.7%)	7 (6.2%)	
Education				
Compulsory school	4 (16.0%)	11 (13.3%)	15 (13.9%)	0.875 ^2^
Apprenticeship	8 (32.0%)	28 (33.7%)	36 (33.3%)	
Technical or commercial school	4 (16.0%)	20 (24.1%)	24 (22.2%)	
High school degree	6 (24.0%)	14 (16.9%)	20 (18.5%)	
(Technical) college/university	3 (12.0%)	10 (12.0%)	13 (12.0%)	
Monthly net household income (€)				
Less than 1000	1 (4.17%)	28 (40.0%)	29 (30.9%)	0.001 ^2^
1001 to 1500	4 (16.7%)	20 (28.6%)	24 (25.5%)	
1501 to 2000	10 (41.7%)	12 (17.1%)	22 (23.4%)	
2001 to 2500	6 (25.0%)	7 (10.0%)	13 (13.8%)	
2501 to 4000	3 (12.5%)	3 (4.3%)	6 (6.4%)	
Population at place of residence				
Less than 2000	4 (15.4%)	3 (3.7%)	7 (6.5%)	0.187 ^2^
2001 to 5000	7 (26.9%)	36 (44.4%)	43 (40.2%)	
5001 to 10,000	3 (11.5%)	14 (17.3%)	17 (15.9%)	
10,001 to 15,000	5 (19.2%)	13 (16.0%)	18 (16.8%)	
15,001 to 20,000	3 (11.5%)	4 (4.94%)	7 (6.5%)	
20,001 or more	4 (15.4%)	11 (13.6%)	15 (14.0%)	
Employment				
Full time	13 (50.0%)	10 (11.9%)	23 (20.9%)	<0.001 ^2^
Part time	0 (0%)	22 (26.2%)	22 (20.0%)	
Retired	12 (46.2%)	42 (50.0%)	54 (49.1%)	
Not working	1 (3.85%)	10 (11.9%)	11 (10.0%)	
Cohabitation with PwD				
No	10 (37.0%)	34 (39.5%)	44 (38.9%)	0.995 ^2^
Yes	17 (63.0%)	52 (60.5%)	69 (61.1%)	
Burden level (BSFC-s)				
Mean (SD)	11.6 (4.77)	14.2 (8.20)	13.6 (7.63)	0.055 ^1^
Disruptive behaviour (NOSGER scale)				
Mean (SD)	10.9 (2.99)	9.97 (3.05)	10.2 (3.04)	0.249 ^1^
Barthel Index				
Mean (SD)	76.0 (19.9)	63.5 (26.5)	66.5 (25.5)	0.026 ^1^
Median [Min, Max]	75.0 [40.0, 100]	65.0 [0, 100]	70.0 [0, 100]	
Quality of Life (EUROHIS-QOL)				
Mean (SD)	3.73 (0.555)	3.69 (0.661)	3.70 (0.637)	0.779 ^1^
Median [Min, Max]	3.63 [2.75, 4.75]	3.75 [2.13, 5.00]	3.75 [2.13, 5.00]	
Caregiving duration (years)				
Mean (SD)	2.72 (1.53)	5.30 (4.38)	4.67 (4.03)	<0.001 ^1^
Weekly caregiving (days)				
Mean (SD)	6.13 (1.78)	5.56 (2.10)	5.69 (2.04)	0.196 ^1^
Daily caregiving (hours)				
Mean (SD)	4.88 (5.25)	8.39 (8.43)	7.54 (7.91)	0.016 ^1^
Home nursing service usage				
No	6 (22.2%)	35 (44.9%)	41 (39.0%)	0.064 ^2^
Yes	21 (77.8%)	43 (55.1%)	64 (61.0%)	

BSFC-s = Burden Scale for Family Caregivers—short version; Max = maximum; Min = minimum, NOSGER = Nurses’ Observation Scale for Geriatric Patients, PwD = person with dementia, SD = standard deviation. ^1^ = *t*-test for independent samples, ^2^ = chi-sqare-test.

## Data Availability

Data are available on reasonable request to D.S.-S as the corresponding author. Trial Registration: The study is registered in the German Clinical Trials Register (ID: DRKS00014749).

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
