# Peer review of "Caring for a Relative with Dementia: Determinants and Gender Differences of Caregiver Burden in the Rural Setting"

_brainsci, 2021, doi:10.3390/brainsci11111511_

Round 1
Reviewer 1 Report
The topic on gender differences in caregiving burden at rural setting is rare, but this paper did not give a fair picture of the situation at rural-urban differences, as well as the gender differences in rural setting. Given the small sample size, the authors adopted univariate analysis and path model to illustrate the possible mediating impact on gender differences and caring burden. However, the writing appears to be tedious and lengthy.
Introduction section:
- It stated “the risk of caregiver burden associated with caring for a PwD is higher compared with caring for a person without dementia”, please clarify the comparison nature (same age group? Any gender difference? etc..).
- Please provide more specific detail on the gender differences in rural setting. Also, the rural-urban differences.
Methodology section:
- Please add a figure to describe the Pearlin´s model and state the instruments you used to measure each dimension.
- Please fix the issue (see Error! Reference source not found).
- Please provide Cronbach’s alpha for all the instruments you used.
- Figure 1 is abundant
- Please give the chi-squared test in table 1 to show if there is a significant gender difference.
- Figure 6 shows that only disruptive behaviour has a direct significant effect on BFSC-s, but not cohabitation.
- Try interacting with the significant mediators and interacting with gender further on the BFSC-s?
Conclusion section: The conclusion is fairly lengthy. Please highlight the study's findings in the beginning.
Reviewer 2 Report
The introduction on the burden of family caring for older people with dementia has been written very general. You need to discuss in here the exact roles of family caregivers in the provision of care to such patients and how it can cause increased caring loads. Studies have been published in which various aspects of the load have been discussed and presented.
As for the method, please go through the equator website, find the equator appropriate to your cross-sectional study and ensure that all required details have been presented:
The Strengthening the Reporting of Observational Studies in Epidemiology (STROBE) Statement: guidelines for reporting observational studies | The EQUATOR Network (equator-network.org)
With regard to sampling, please give information about the probable sample size, the estimated sample size for this study, any sampling formula used, the withdrawal numbers etc.
Lines 144-154, please add citations to those instruments used for data collection.
I can see that you have used a couple of data collection tools in this research. Theerfore, you should improve the research aim and add objectives to cover the logic for the use of all these data collection tools.
A table or figure really can help depict the data collection process and instruments.
No ethical considerations have been presented.
Please describe how the data have been practically collected and when etc?
Lines 240-241, where can it be found in the figure or table?
Round 2
Reviewer 2 Report
Your response to my question about the sampling, recruitment, withdrawal etc should added to the limitations section at the end of this article. Given that your samples are not representative, weaknesses should be recognised in there.
